# Measures and Penalties for Animal Welfare Violations at German Abattoirs: A Compilation of Current Recommendations and Practices

**DOI:** 10.3390/ani13182916

**Published:** 2023-09-14

**Authors:** Stephanie Janet Schneidewind, Diana Meemken, Susann Langforth

**Affiliations:** Working Group Meat Hygiene, Institute of Food Safety and Food Hygiene, School of Veterinary Medicine, Freie Universität Berlin, 14163 Berlin, Germany; diana.meemken@fu-berlin.de (D.M.); susann.langforth@fu-berlin.de (S.L.)

**Keywords:** slaughterhouse, animal cruelty, breaches of animal welfare, transgressions of animal protection laws

## Abstract

**Simple Summary:**

In Germany, animal welfare during preslaughter handling and slaughter is governed by national laws and European regulations. However, animal welfare violations still occur at abattoirs, which is an ethical and economic problem. This article describes how our group of research associates investigated which specific measures and penalties responsible authorities consider appropriate for 40 different animal welfare violations at German abattoirs. Past measures and fines are described for some violations. The aim is to provide insight into the status quo, so that flaws in law enforcement can be identified. An online survey, semi-structured interviews, and a virtual colloquium with official veterinarians were conducted. Additionally, relevant judicial decisions from Germany were collected and summarised. Legal professionals were consulted for assistance. Our findings were compiled into a list of measures and penalties. This project is a first step towards achieving a more consistent and standardised enforcement of the German Animal Welfare Act, and thus towards reducing the occurrence of animal welfare violations at German abattoirs.

**Abstract:**

Protecting animal welfare during preslaughter handling and slaughter is an important ethical concern with growing importance to consumers. However, animal welfare violations in abattoirs remain a serious problem, and the enforcement of relevant laws and regulations is often inadequate. This study investigated and compiled the measures and penalties which official veterinarians (OVs) consider appropriate for different animal welfare violations at German abattoirs, including ranges for fines. Additionally, information regarding which measures were taken in past cases, including past procedural outcomes (e.g., judicial decisions and regulatory animal welfare orders in Germany), were gathered and summarised. The aim is to provide insights into the status quo, so that flaws in law enforcement (e.g., imposing low penalties or not filing a criminal complaint when necessary) can be identified in a future study. To achieve this, the following five steps were utilised: acquiring relevant judicial decisions; conducting an anonymous online survey among German OVs; conducting semi-structured interviews with OVs; conducting a virtual colloquium with OVs; and consulting lawyers. Measures and penalties for violations of 40 relevant and frequent different provisions of the German Ordinance on the Protection of Animals in connection with Slaughter or Killing (TierSchlV), in conjunction with the Regulation (EC) No 1099/2009 and the German Animal Welfare Act, were gathered. The findings were compiled into a list of measures and penalties, which contains a separate table for all 40 violations, entailing an overview of the following information: citation(s) of legal/regulatory requirements to protect animals at the time of killing/slaughter; citation(s) of relevant regulatory and criminal penalties; special measures/penalties considered appropriate for the first and repeated offence by OVs; and information on penalties listed in judicial decisions of past similar cases. This initiative is a step towards achieving a reduction of animal welfare violations at German abattoirs.

## 1. Introduction

In Germany, the treatment of animals in abattoirs is regulated by Regulation (EC) No 1099/2009 [1], the German Animal Welfare Act (Tierschutzgesetz) [2], and the German Ordinance on the Protection of Animals in Connection with Slaughter or Killing (Tierschutzschlacht-Verordnung) [3]. According to Regulation (EC) No 1099/2009, abattoir operators are principally responsible for ensuring compliance with animal welfare laws and regulations at the abattoir. Official veterinarians (OVs) check compliance with the Animal Welfare Act and the German Ordinance on the Protection of Animals in connection with Slaughter or Killing in conjunction with Regulation (EC) No 1099/2009, and, if necessary, take the actions required to remedy deficiencies. Section 18 of the German Animal Welfare Act provides vague guidelines for penalties when violations of animal welfare laws and regulations in abattoirs (which will be referred to as “animal welfare violations” from now on) are committed: A regulatory fine of up to EUR 5000.00 or EUR 25,000.00, depending on the violation, can be issued for an offense against the German Ordinance on the Protection of Animals in connection with Slaughter or Killing in conjunction with Regulation (EC) No 1099/2009. Section 17 of the German Animal Welfare Act states that anyone who inflicts considerable pain or suffering out of cruelty and/or persistent or repeated severe pain or suffering on a vertebrate shall be liable to up to three years imprisonment or a criminal fine. 

Animal welfare violations present an important ethical issue, seeing as they inflict avoidable pain, distress, and suffering upon slaughter animals [4]. Furthermore, poor animal welfare during preslaughter and slaughter impacts meat quality negatively [5,6,7], causes economic losses [4], and conflicts with consumer demand for high standards of animal welfare [8]. Violations of animal welfare standards during slaughter are a pervasive global issue, exhibiting varying degrees of occurrence [9,10]. A study on animal welfare during slaughter in Portugal, Italy, Finland, Brazil, and Spain reported signs of recovery after stunning ranging from 0% to 90% in pigs [11]. According to a scientific study published by Reymann in 2016, animal welfare violations occurred during preslaughter handling, stunning, and exsanguination in all of the Bavarian abattoirs audited [12]. A separate report on the inspections of Bavarian abattoirs in 2014 and 2015 found that more than 50% of the abattoirs inspected showed significant deficiencies, including animal welfare violations [13]. The identified deficiencies encompassed the following: the utilisation of outdated equipment, which resulted in ineffective stunning of slaughter animals, as well as structural shortcomings and a lack of adequate water supply [13,14]. The findings reported in the literature have been corroborated by undercover investigations by animal welfare organisations, which receive a lot of media attention [15]. Articles describing (and sometimes showing) footage of violations of Regulation (EC) No 1099/2009 frequently make headlines in European media (e.g., beating animals or slaughtering ineffectively stunned animals) [16,17,18,19,20]. Additionally, publications describe that there is a deficit in the enforcement of animal welfare laws and regulations at German abattoirs [21]. In general, penalties are particularly low for violations committed against farm animals [22]. Thilo reported that there is no correlation between the severity of an offence and the outcome of the proceedings, and that sanctioning animal welfare violations is inconsistent and often omitted [23]. Lawyer Hahn commented on the court decision of the Higher Regional Court of Frankfurt am Main (Ref. 2 Ss 194/20), which described that deficiencies were tolerated in a German pig abattoir by responsible authorities over years prior to the court sentence, as follows: “German criminal law regarding animal protection exists primarily on paper. Especially in the case of farm animals, there is hardly any actual prosecution of such offences. This also—and especially—applies to animal welfare violations in abattoirs” [24]. When violations are sanctioned, the penalties are not dissuasive. Rather, there is a certain degree of tolerance towards the mistreatment of farmed animals [25]. Consequences for common violations vary greatly, and sometimes there are no consequences despite official veterinarians documenting and reporting violations [25]. Literature describes that responsible employees are subjected to a verbal instruction and/or an administrative proceeding is initiated in the case of a relatively mild violation [26]. In the case of repeated or significant deficiencies, a regulatory offence or criminal proceedings may be initiated. In cases of severe violation, proceedings may result in the suspension or withdrawal of the establishment’s license [26]. Publications on this subject, however, do not categorize different violations according to severity. Contrary to other countries (e.g., Canada [27]), there are no accessible guidelines for the distinction for mild and serious violations at the slaughterhouse. Hahn and Kari described the following violation to be “severe”: An employee deliberately bends the tail of a cow and subjected it electric shocks using an unauthorized electric stunning device [8]. The status quo regarding law enforcement in German abattoirs (e.g., sanctions or recommendations for measures and penalties) has not yet been formally investigated. This is also true for other European countries. Various articles focus on the assessment of animal welfare at the abattoir, such as developing a scoring tool for the risk of non-compliance with animal welfare regulations [28]. Other sources provide tips on how to reduce stress and suffering and ease handling of slaughter animals [29]. Some describe corrective actions for problems, but do not include penalties [30]. The German guidelines for the implementation of Regulation (EC) No 1099/2009 provides checklists and general advice for addressing deficits, but it does not describe specific measures and penalties suitable for different concrete violations. Occasionally, sources provide ideas for specific measures, such as hanging up banners with clear instructions to remind employees of which behaviours or tools are not accepted [31], but this does not reveal any information on the status quo of how OVs enforce animal protection laws or what they deem appropriate. This topic is worthy of investigation, seeing as German OVs are criticisedcriticised for not taking enough action in cases of animal welfare violations [32]. 

In total, this project developed a list of the measures and penalties that participating OVs evaluated as appropriate for relevant and frequent violations of 40 provisions of the German Ordinance on the Protection of Animals in connection with Slaughter or Killing, in conjunction with the Council Regulation (EC) No 1099/2009 and the German Animal Welfare Act. This information is crucial for identifying shortcomings in law enforcement and exploring ways to improve animal protection at abattoirs. This paper will describe the methodology and outcome of the different steps used to complete this research project. It is important to note that the measures and penalties collected here represent recommendations given by responsible authorities, which demonstrate the status quo of how violations are handled by responsible authorities, rather than presenting how such violations should and must be handled. In the future, experienced legal professionals specialised in administrative, regulatory, and criminal law at abattoirs should thoroughly review and refine these measures and penalties to establish helpful guidelines. This initiative shows great potential in promoting a more consistent and standardised enforcement of the German Animal Welfare Act, thereby potentially reducing the occurrence of animal welfare violations at German abattoirs.

## 2. Materials and Methods

The list of measures and penalties (including those which OVs recommended and those implemented in the past) was compiled using the following five “steps”: acquiring all obtainable relevant judicial decisions from Germany; conducting an anonymous online survey among German OVs; conducting semi-structured interviews with OVs; conducting a virtual colloquium with OVs; and consulting lawyers throughout the course of the project. This project focused on violations at the abattoir, meaning that violations which can occur during transport to the abattoir were not addressed. 

Measures and penalties considered appropriate by OVs, in addition to current practices in animal welfare law enforcement (e.g., judicial decisions and past measures described by survey participants), were compiled for 40 relevant and frequent possible animal welfare violations. The violations addressed were ones which occur during preslaughter handling, stunning, and exsanguination. The violations present regulatory offences according to Section 16 of the German Ordinance on the Protection of Animals in connection with Slaughter or Killing. A framework for a regulatory fine (which can also be applicable for a periodic penalty payment) was included for regulatory offences, which represents the median values of the ranges for fines considered appropriate by our survey participants. For the definition of a regulatory offence and a crime, please see Appendix A. For each possible violation, a table which presents an overview of the relevant section of an Act/Regulation (Regulation (EC) No 1099/2009; German Animal Welfare Act; German Ordinance on the Protection of Animals in connection with Slaughter or Killing); measures and penalties considered appropriate for the first and repeated violation; and details of the outcomes of judicial decisions for past similar cases were included. The final document containing the previously described tables for 40 different animal welfare violations at abattoirs was translated from German into English and can be found in the list of measures and penalties (Appendix A). The violations described fall into one of the following categories: (1) Violence against slaughter animals and/or use of prohibited driving aids; (2) inadequate housing/husbandry of animals in lairage; (3) restraining, stunning, and bleeding animals in a manner which violates animal welfare standards; (4) inappropriate handling of ill/injured animals; and (5) individual employees performing tasks relevant to handling/slaughtering animals without an appropriate certificate of competence. 

### 2.1. Acquisition of Relevant Judicial Decisions

Judicial decisions regarding animal welfare violations during preslaughter handling, stunning, and exsanguination were gathered in order to collect information on how courts handled past violations at abattoirs. In the beginning, legal databases (OpenJur [33], Juris [34], beck-online [35]) were used to retrieve relevant available judicial decisions from Germany. Violations occurring during the transport of animals to the abattoir were not analysed, since our study focused exclusively on violations committed at the abattoir. Since only very few judicial decisions could be obtained using the previously mentioned databases (N = 5), media reports on animal welfare violations were systematically researched. For this purpose, an online list of all approved abattoirs (from the year 2006) was used to enter the following search terms into Google © (in German): Abattoir X + Animal Welfare + Breach/Violation/Transgression. “X” refers to a city in which an abattoir was located according to the list. A list from 2006 was used because it was the most recent list available. Additionally, this enabled us to find violations which may have occurred in abattoirs before they closed between 2006 and our investigation in 2021. Relevant information available in the article was recorded (e.g., the species affected; the reference number; the location of the abattoir). The press offices of a total of 33 departments of public prosecution were contacted via e-mail to inquire about 33 different animal welfare violations and asked to provide the corresponding judicial decision for scientific purposes. The judicial decisions received were then summarised and included in the overview, which can be found in Appendix A, in a column specifically for relevant judicial decisions and notes. 

### 2.2. Conducting an Anonymous Online Survey among OVs

An anonymous online survey was conducted in order to gather information on measures and penalties for specific animal welfare violations at abattoirs from persons with relevant work experience. The study was conducted in accordance with the Declaration of Helsinki, and approved by the Ethics Committee of Freie Universität Berlin (protocol code ZEA-Nr. 2022-007; date of approval: 11 April 2022). The target audience for the survey were OVs and others entrusted with enforcing animal welfare laws and regulations in German abattoirs (e.g., animal welfare officers). Participation in the survey was voluntary and there were no mandatory questions. The survey included questions regarding frequent cases of violations of Article 16 of the German Ordinance on the Protection of Animals in connection with Slaughter or Killing, which were identified through available literature, judicial decisions, media reports, and interviews with OVs. Altogether, 22 persons tested a mock-up to validate the survey. Suggestions for improvement were incorporated, leading to the final survey. The final survey included 22 constructed but realistic cases of animal welfare violations. In this section, these will be referred to as “cases”. Most cases (18/22) were at least regulatory offences according to Article 16 of the German Ordinance on the Protection of Animals in connection with Slaughter or Killing and the Regulation (EC) No 1099/2009, but in some cases could also be considered crimes due to the prolonged pain and/or suffering inflicted on the animal. The remaining cases also described situations which inflicted considerable pain and/or suffering on the animals, but a description that these specific violations constitute a regulatory offence cannot be found in German or European regulations. The survey was programmed using the online survey tool LimeSurvey, Version 3.28.21. In the beginning, the survey contained questions about the participants’ profession in order to get an impression of their experience and competence. This later allowed screening in terms of a participants’ eligibility for being included in the analysis. Suggestions for improvement were incorporated, leading to the final survey. Overall, the final questions and cases fell into one of seven question groups: questions regarding the participant’s professional experiences (10 questions); violence against slaughter animals and/or the use of prohibited driving aids (8 cases); inadequate housing/husbandry of animals in lairage (3 cases); restraining, stunning, and bleeding animals in a manner which violates animal welfare standards (8 cases); inappropriate handling of ill/injured animals (2 cases); employees performing tasks without an appropriate certificate of competence (1 case); and one question regarding the most common violations in the participant’s work environment (see Appendix A for the survey in English and Appendix A for the survey in German). 

The answers were provided in multiple-choice format. Additionally, a comment box without a word limit was provided for each question. For every case in the survey, the same four questions were asked. The first question addressed what measures/penalties the participants felt would be appropriate if this animal welfare violation occurred (hypothetically). Appropriate measures could be selected for the first and repeated violation, and a range for an appropriate regulatory fine (in EUR) could be provided for the first and repeated violation, if the participant selected that a regulatory fine is an appropriate penalty (e.g., “EUR 200–EUR 500” for the first violation and “EUR 400.00–EUR 1000.00” for the repeated violation). The second question inquired about the occurrence of a similar case in the participant’s work environment in the past three years. Information could be given as to whether a similar case to the violation described had occurred and what measures/penalties were taken or ordered. The question inquired about the past three years, because the measures taken (including penalties) should be relatively recent, in order to reflect the current status quo. The case and/or the measures taken could be described in as much detail as desired using the comment box. Quantitative information on the regulatory fine, the warning fee, or a criminal fine (after a court case) was requested explicitly. The third question dealt with whether, from the point of view of the participant, other/additional measures would be necessary in this case. Other comments on the case could also be made here (e.g., whether the manager or the employee should be held accountable). 

The survey was available online for two months from 1 March 2022 to 30 April 2022. Participants were recruited by sending an e-mail including the link for the survey to all veterinary authorities in Germany (N = 431). Calls for participation were also printed in two articles in specialised journals for German-speaking veterinarians and participants in an online conference (attended by approximately 450 people) were informed about the survey before it started. 

The data collected were analysed statistically using IBM ^®^ SPSS Statistics Version 27 (SPSS, Inc., Chicago, IL, USA) and Microsoft Excel 2019 ©. A measure/penalty was included in the respective table for the individual violation (which can be found in Appendix A) if over 50.0% of participants evaluated a specific measure as appropriate. The tables in the list of measures and penalties summarise which measures and penalties project participants assessed as appropriate for a specific breach of animal welfare regulations for the first and repeated violation. Additional measures mentioned in the comments section were included if there were comprehensible reasons, such as being based on legal norms or professional experience. Descriptive statistics (mean, median, minimum, maximum, range, and standard deviation) of all suggested values for appropriate regulatory fines were calculated using SPSS. The ranges of appropriate fines and information on past measures (e.g., how high a regulatory fine was) were listed in the list of measures and penalties (Appendix A). 

### 2.3. Conducting Semi-Structured Interviews with OVs

Semi-structured interviews with six experienced OVs were conducted to gather information on different aspects of animal welfare violations at abattoirs. These OVs were personal contacts of the project’s working group. The aim was to gain deeper insights into current practise of measures taken in response to animal welfare violations. Interviews were conducted either over the phone or via the web conferencing platform Cisco Webex. Questions were prepared in advance (e.g., Which kind of animal rights violations occur most frequently in your work environment? Which procedures do you follow when an incident occurs? How is the amount of a regulatory fine decided? What challenges do you face regarding law enforcement?). A full set of the survey questions can be found in Appendix A. However, interviewees were encouraged to share any relevant information which they believed might be beneficial for the project. 

### 2.4. Conducting a Virtual Colloquium with OVs

In July 2022, a virtual colloquium was conducted on two consecutive days via Cisco Webex to discuss how the measures and penalties proposed by participants of the online-survey should be improved, so that they truly represented the actions which the OV would consider appropriate for different violations. This gave OVs in all German federal states an opportunity to participate. Four hours per day were allocated to ensure enough time for all possible animal welfare violations. An invitation to participate was sent to all veterinary authorities in Germany via e-mail. People who registered to participate received the draft before the colloquium by e-mail, in order to incorporate suggestions for improvements/alterations in advance. During the event, the draft of the Word© document was shared on screen, allowing all attendees to see what changes were made to the document in real time. The aim of this event was to allow participants to discuss suggestions for changes to the measures and penalties proposed, and find a consensus in the end. Requests for modifications could either be provided verbally or in the chat. Overall, 40 possible violations of animal welfare regulations were discussed individually, corresponding to each violation, which can now be found in the list of measures and penalties (in Appendix A). A vote was taken on important and complex issues, with participants voting in the chat or “raising their virtual hand” (a user-function on Cisco Webex). 

### 2.5. Consultation of Lawyers throughout the Course of the Project

Throughout the course of the project, two lawyers were available to provide guidance. They were involved in order to ensure the correct use of legal terminology when designing the online survey. Additionally, the two lawyers supported the acquisition and analysis of judicial decisions. The lawyers involved were research associates. One is a Visiting Professor who works inter alia on questions of German Public Law and European Law. The other lawyer is a research associate in criminal law, who has published articles on animal welfare violations at the abattoir. The measures and penalties proposed were revised from a legal perspective by both lawyers. 

## 3. Results

This project compiled a list of recommendations for measures and penalties for the 40 violations which can be found in Table 1. Further information can be found in Appendix A. 

### 3.1. Acquisition of Relevant Judicial Decisions

By the end of the project, a total of 16 German judicial decisions from the years between 2015 and 2022 were obtained, which provided valuable insights into the enforcement of animal welfare laws and regulations at abattoirs. There were 20 negative responses to the requests for specific judicial decisions, citing various reasons such as the absence of a reference number, concerns regarding data privacy, or because the court case was still pending. Violations included in the judicial decisions were one or more of the following (listed in descending order of frequency): exceeding the maximum time allowed to pass between stunning and bleeding without a certificate of exemption (in accordance with Section 13 Paragraph 2 of the German Ordinance on the Protection of Animals in connection with Slaughter or Killing); an illegal use of devices which administer electric shocks, kicking/beating animals, exsanguination of ineffectively stunned animals (e.g., an animal which shows signs of consciousness such as spontaneous blinking/directed eye movements/reactions to touch), dragging animals which are too weak or injured to walk on their own with painful driving aids (e.g., a winch), a failure to supply drinking water to animals as required by relevant regulations, and not milking lactating dairy cattle every twelve hours. Several violations of the German Ordinance on the Protection of Animals in connection with Slaughter or Killing and the German Animal Welfare Act are often judged at the same time. Therefore, it was not always possible to provide information regarding which violations lead to which procedural outcomes. However, in many cases, information could be summarised and incorporated into the overview of measures and penalties for different violations, citing a specific fine for a specific violation. These summaries can be found in the respective table for a specific violation in Appendix A. 

### 3.2. Conducting an Anonymous Online Survey among OVs on Animal Welfare Violations at German Abattoirs

#### 3.2.1. Participants 

In total, 312 persons started the survey. Most participants left a varying number of questions unanswered. Altogether, 204 participants (65.4%) ended the survey after the section regarding their profession (questions designed to screen participants). For a participant’s responses to be included in the analysis, questions corresponding to at least one case had to be answered. The entire survey was completed by 66 OVs, meaning that N varied between 66 and 108 participants. Information about the participants in the online-survey can be found in Table 2, Table 3, Table 4 and Table 5. 

#### 3.2.2. Survey Results

The survey gathered assessments on appropriate measures and penalties for hypothetical violations, as well as information on measures taken if violations occurred in the participants’ work environment in the past. The responses corresponding to appropriate measures in the event of a first and repeated violation showed that participants were in favour of intervening on-site (if possible) rather than punishing violations. Measures with greater financial or work-related consequences, e.g., losing the certificate of competence, tended to be evaluated as appropriate more frequently in the event of a repeated violation. The most common actions identified as appropriate were the following: Intervening and immediately correcting the incorrect working method/technique; informing the supervisor and/or animal welfare officer; conducting an oral briefing with the responsible person(s) (and/or request the animal welfare officers and/or the supervisors of the responsible person(s) to conduct an oral briefing); and more frequent inspections in the affected area of the abattoir. These measures were identified as “basic measures” for an animal welfare violation at the abattoir, seeing as they were favoured by over 50.0% of participants in most cases, as demonstrated in the case shown in Figure 1 and Figure 2. Two cases were outliers regarding favoured measures and penalties. The first outlier is a case in which an animal is not stunned prior to exsanguination for the purpose of slaughter, without a respective official exemption permit (e.g., for the purpose of Halal/Kosher butchering). The second outlier is a case in which a downer cattle is dragged out of a vehicle and into a slaughterhouse instead of humanely killing it according to German regulations. In this case, participants evaluated measures with greater consequences as appropriate for the first violation. Listing these measures as “basic measures” despite the two outliers was agreed on in the virtual colloquium. 

The data for all cases (N = 22) were analysed separately. Fines were calculated for violations which either constitute a regulatory offense according to Section 16 of the German Ordinance on the Protection of Animals in connection with Slaughter or Killing, or when more than 50.0% of the participants agreed that a regulatory fine was appropriate. All the ranges calculated can be found in the list of measures and fines in the Appendix A. 

As can be seen in Figure 1 and Figure 2, participants’ assessments regarding which measures/penalties would be appropriate for a specific animal welfare violation varied greatly. Measures considered appropriate by over 50.0% of participants were included in the first draft of the list. On average, 18.9% of participants suggested a regulatory fine after the first violation and 45.5% did so for the repeated violation, even though 18/22 cases were at least regulatory offences according to German law. Additionally, the results suggested that many OVs do not file a criminal complaint when this is imperative. For example, many OVs suggested that dragging a downer cattle off of the transportation vehicle instead of humanely killing it on the spot is a regulatory offence, while this should be prosecuted as a crime. This was determined in an animal welfare conference that took place in Munich in March 2023, where this specific case was discussed.

### 3.3. Conducting Semi-Structured Interviews with OVs

Interviews revealed frequent animal welfare violations in the working environment of the individual interviewee. The most commonly mentioned violations were similar to the violations mentioned in literature (e.g., illegal use of electric prods, or other devices which administer electric shocks, and exceeding the time limit allowed between stunning and bleeding). In some cases, OVs described systemic problems. Several OVs reported that, despite consistently documenting and reporting violations to veterinary authorities, measures were not taken and penalties not imposed in many cases. In addition, several interviewees described that, in many cases, they did not receive feedback on the course or the outcome of the procedure. This is coupled with a lack of support from management in veterinary authorities when violations were reported. Rather, abattoirs faced no consequences following violations. Furthermore, OVs described that they receive threats in abattoirs in response to informing them that they will report a deficit/an animal welfare violation. As a result, official staff may deliberately choose not to report a violation, because they fear personal and/or professional consequences. It is also important to note that OVs may issue measures directed to the future, such as an administrative proceeding based on Section 16a of the German Animal Welfare Act. 

### 3.4. Conducting a Virtual Colloquium with OVs

A virtual colloquium was conducted with OVs to discuss the measures and penalties deemed appropriate in the online survey. The aim of this discussion was to evaluate the survey results in a practical context, and to improve them, so that they could be used as guidelines. Between 20 and 30 well-experienced OVs participated in the virtual colloquium on both days. The number varied because some participants entered the conference late and/or left early on both days. Participants of the virtual colloquium agreed that the median values should be used for the framework of fines because the median values do not consider the extreme values recommended by few participants. Furthermore, it was agreed that the values for the ranges of the regulatory fines for repeated violations should correlate to double the median values of the first violation. In some cases, the values were not just considered appropriate for a regulatory fine, but also for a periodic penalty payment. Additionally, the virtual colloquium clarified that animal welfare orders based on Section 16a of the German Animal Welfare Act (TierSchG) are frequently issued in order to tackle deficiencies in the abattoir in order to prevent them in the future. “Animal welfare orders” are administrative orders by the competent authority that can legally oblige an addressee to take certain actions/measures. This particular measure was assessed as appropriate rather infrequently in the online survey (on average by 10.7% of participants in the case of the first violation and by 29.3% of participants in the case of a repeated violation). Specific measures for each violation were added to the document which provides an overview of measures and penalties considered appropriate, if a participant described a measure to be appropriate based on legal norms and/or relevant professional experience. As previously mentioned, there were 22 cases in the survey and 40 cases intended for the final overview of measures and penalties deemed appropriate. Various cases are comparable in terms of severity and which measures/penalties are appropriate when they occur. We discussed which cases were comparable to the cases in our survey and then adapted the appropriate measures and penalties to the individual case throughout the course of the virtual colloquium. Thereby, a list of measures and penalties deemed appropriate was completed for 40 violations of animal welfare laws. The 40 violations discussed can be found in Table 1. 

### 3.5. Consultation of Lawyers throughout the Course of the Project

The measures and penalties considered appropriate by OVs were revised by two lawyers, who provided essential feedback in terms of correct legal terminology. There were no contradictory suggestions between the two lawyers. 

## 4. Discussion

The objective of this research project was to compile a list of measures and penalties considered appropriate by veterinary and legal authorities for 40 animal welfare violations at German abattoirs. When applicable, past sanctions were summarized, including a description of the violation and the number of daily rates. The results were achieved by carrying out five important steps. The steps which included OVs were the following: An online survey, semi-structured interviews, and a virtual colloquium. Additionally, legal professionals were consulted, and relevant judicial decisions from Germany were collected and summarised. This project is the first step towards providing guidelines for OVs responsible for overseeing animal welfare in abattoirs. Providing such guidelines is important, because OVs bear a great responsibility to ensure that animal welfare offenses are appropriately punished [36]. In Germany, OVs are subjected to a “duty of care and protection” (Garantenpflicht), as described in Section 16a of the Animal Welfare Act [37]. If the case is not handed over to the public prosecutor’s office, criminal liability for obstruction of justice by omission will be considered (Sections 258, 258a, 13 of the German Criminal Code) [38]. The responsible OVs can be held criminally liable for omission, either as an accomplice or for aiding and abetting under Section 27 of the German Criminal Code.

As for interpreting the results of this study, lawyers and OVs with experience in law enforcement in abattoirs should carefully review the list and identify whether the measures and fines suggested are adequate. However, the results of the online survey on their own suggest that animal protection laws at the abattoir are under-enforced by some OVs. Most of the violations were at least regulatory offences, but the average amount of participants suggesting an initiation of a regulatory offence proceeding after the first violation was only 18.9%. For the repeated violation, this percentage increased to 45.5%. We would have expected the percentage of participants recommending regulatory fines in the violations depicted in Figure 1 and Figure 2 to be closer to 100.0%, seeing as Section 16 lists this violation as a regulatory offence which should be fined. Additionally, the results of the online survey showed that many OVs would not file a criminal complaint in cases of significant pain or prolonged suffering. For example, many OVs suggested that dragging a downer cattle off of the transportation vehicle instead of humanely killing it on the spot is a regulatory offence, even though this should be prosecuted as a crime due to the significant and prolonged pain it inflicts on the animal. Perhaps this is due to the lack of support that many OVs reportedly receive from the department head of their office, or these results suggest that OVs may need further training in identifying animal welfare violations and implementing the measures and fines the law requires them to. However, it is also important to note that the results of the survey contrasted with the results of the virtual colloquium. During the virtual colloquium, participants suggested initiating measures aimed at addressing the specific underlying issues causing an animal welfare violation. This often involved an animal welfare order based on Section 16a of the German Animal Welfare Act. This creates the impression that the animal welfare order based on Section 16a is a more promising solution than punishment for less severe cases. This approach focuses on solving problems rather than punishing abattoir employees and/or operators. As for the fines listed in judicial decisions, it is known that most public prosecutors have little experience with animal welfare violations of farm animals and thus may suggest rather low fines [25]. However, on the other hand, some lawyers would suggest very high sanctions, as demonstrated by Jens Bülte’s suggestion to increase the prison sentence for crimes committed against animals from three to five years [39]. When penalising a violation, the individual level of pain, suffering and/or harm inflicted on the animal(s) must be considered. Moreover, the income and personal circumstances, in addition to whether it was their first or repeated violation, must be taken into account. Additionally, a decision must be made as to whether the general behaviour of the persons accused indicates empathy for the animals affected, or whether animals merely have the status of goods. This may, to some extent, explain why the participants’ responses regarding appropriate measures and penalties for specific animal welfare cases varied so much in the survey (as exemplified by Figure 1 and Figure 2). Further reasons, such as the lack of guidelines, are likely to contribute to the heterogeneity of participants’ assessments. Whether the regulatory fines and/or periodic penalty payments proposed by participants are too high, too low, or just right, is rather subjective and might depend on the personal experiences gained in the context of a similar past case which occurred in an OV working environment. Therefore, these measures and penalties cannot be used as guidelines for law enforcement at abattoirs, but rather provide insights in the status quo and provide a basis for developing clear guidelines. 

To our knowledge, this project is the first to present insights into which measures and penalties authorities deem(ed) to be appropriate for animal welfare violations in German abattoirs. Given the unprecedented methodology which was applied, the findings can hardly be compared to other publications. Some fines for regulatory offences from the year 2019 were published by the Administrative District of Kassel [40]. For example, the first time an abattoir was caught slaughtering a cow in the last trimester of pregnancy, it was fined with EUR 500.00. When this abattoir recommitted this violation, the fine was EUR 1000.00. This is in line with the results of our study, seeing as the participants in the online colloquium agreed that the fine of the repeated violation should be double the fine of the first violation. Another violation in Kassel described how a sheep was not stunned prior to exsanguination during slaughter. The fine imposed, which was EUR 300.00, is lower to the median of the fines which OVs who participated in our study recommended for this violation (EUR 500.00 to EUR 1000.00). However, it must be noted that details about the case in Kassel are missing. 

The limitations of this study are the following: The data collected as part of the voluntary survey, interviews, and participation in the virtual colloquium cannot be validated. This is a common issue with these methods, which needs to be considered in relation to the benefits gained by using them. Participants were able to share their professional assessments and experiences in an anonymous and/or confidential setting. This is important considering this specific topic, seeing as there is media coverage on the particular subject of OVs not responding to animal welfare violations adequately [41,42]. It is possible that only OVs who are especially motivated to improve the status quo of animal welfare contributed to this project, meaning that they may have suggested harsher measures and penalties. However, the results suggest that participants favoured relatively low fines (see Appendix A), and preferred measures which address the underlying issue of the animal welfare violation. Additionally, there are more judicial decisions than we could gather over the course of this project, meaning that the information presented regarding past cases is incomplete. However, obtaining further judicial decisions was not feasible for several reasons, including pending cases, data protection concerns, and other factors preventing their accessibility. Thus, a complete picture of the status quo of law enforcement could not be provided. Furthermore, this project could not address all the animal welfare violations which can possibly occur in an abattoir. The most relevant and common cases were included. Deficits like construction defects (which often cause animal welfare violations according to Hahn and Kari [8]) were not addressed, seeing as these issues are very individual and hard to assess without very specific details. Causes of animal welfare violations are very individual and vary among abattoirs, which means that the measures and penalties must also be individual. This could not be fully considered and addressed within the framework of this research project. Causes can be one or more of the following: a lack of employee training and/or expertise; negligence; special conditions such as structural and/or constructional deficiencies; time pressure, a high slaughter speed; inadequate infrastructure; and/or economic interests [8,20,21]. These underlying issues remain an important problem in abattoirs, and thus should be addressed in future studies. 

As for future directions, the next step in this project is revising these measures and penalties proposed by OVs into guidelines which clearly state which measures must be taken in the event of specific animal welfare violations. 

These findings can be used to develop clear guidelines regarding which actions can and must be taken for animal welfare violations, and the approach can be adapted and applied in other countries. 

## 5. Conclusions

This project compiled which measures and penalties responsible authorities would consider appropriate and which measures were taken in the past. Legal citations and summaries of relevant court decisions were included as well. This presents the status quo of law enforcement for different cases of animal welfare violations at the abattoir. The online-survey suggested that some OVs do not initiate the measures and penalties which the law requires them to, and thus may need further training in identifying animal welfare violations and taking the necessary actions. Revising these measures and penalties into a list of guidelines and recommendations (which also describe which actions must be taken according to the law) has the potential to promote a more consistent and standardised enforcement of the German Animal Welfare Act. This can contribute to the reduction of the number of animal welfare violations at German abattoirs. 

## Figures and Tables

**Figure 1 animals-13-02916-f001:**
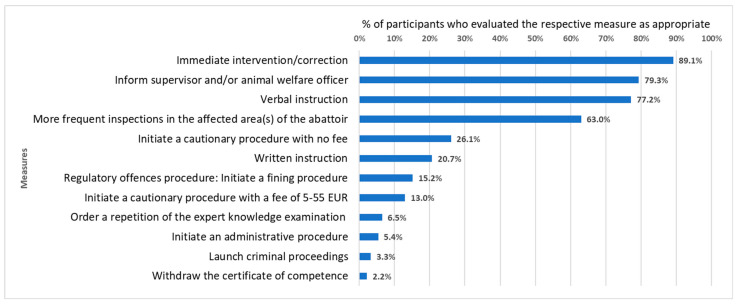
Assessments of appropriate measures regarding a first violation of Article 15(1) in conjunction with Annex III (1.8.)(e) of Regulation (EC) No 1099/2009 (Rotating an animal’s tail by 180°). Multiple answers were possible.

**Figure 2 animals-13-02916-f002:**
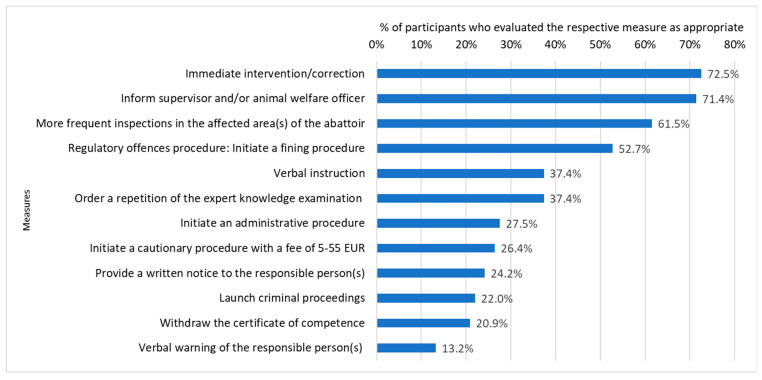
Assessments of appropriate measures regarding a repeated violation of Article 15(1) in conjunction with Annex III (1.8.)(e) of Regulation (EC) No 1099/2009 (Rotating an animal’s tail by 180°). Multiple answers were possible.

**Table 1 animals-13-02916-t001:** The 40 animal welfare violations addressed in this project.

No.	Description of Violation
(1)	Use of instruments which administer electric shocks in contravention of Section 5 of the German Ordinance on the Protection of Animals in Connection with Slaughter or Killing, in conjunction with Annex III(1.9) of Regulation (EC) No 1099/2009, where multiple violations occur simultaneously (e.g., repeated, inadequately spaced out administrations of electric shocks in body regions other than the muscles of the hindquarters).
(2)	Use of instruments which administer electric shocks in contravention of Section 5 of the German Ordinance on the Protection of Animals in Connection with Slaughter or Killing in conjunction with Annex III(1.9 of Regulation (EC) No 1099/2009, whereby one violation occurs (e.g., use of an electric prod on an animal that is too young, but apart from this, the use complies with animal welfare regulations.
(3)	Striking an animal with a driving stick, particularly against a sensitive body part such as the eye, causing harm or distress.
(4)	Kicking an animal.
(5)	Intentionally dropping a door or a gate onto an animal, causing harm or distress.
(6)	Use of a driving tool with a pointed or sharp-edged end to handle or move an animal, causing potential harm or distress.
(7)	Squeezing, rotating or breaking an animal’s tail.
(8)	Intentionally applying pressure to a sensitive body part of an animal, such as the eyes, ears, nose, anus, or genitals, causing undue pain or suffering.
(9)	The act of lifting or pulling an animal by its head, fur, ears, horns, legs, or tail.
(10)	The act of grabbing, carrying or tugging poultry in contravention of animal welfare regulations (e.g., by only one wing, the neck, head, tail, wing tips or plumage).
(11)	Forcing animals to move by shouting, or by exhibiting aggressive behaviour.
(12)	Improper use of a herding board or rattle paddle.
(13)	Animals are obstructed or prevented from moving in the required direction by obstacles or structural defects, or these issues create the possibility of escape.
(14)	The permanent unevenness of the ground, such as holes, can compromise the surefootedness of animals walking on floors (e.g., passageways and holding pens).
(15)	Animals do not have access to an adequate supply of drinking water.
(16)	Animals arriving at the abattoir in containers (e.g., poultry) are not provided with drinking water despite not being sent to slaughter within two hours upon arrival.
(17)	Animals not slaughtered within six hours after arriving at the abattoir are not provided with appropriate or sufficient feed, or there are not enough troughs, or there isn’t a sufficient trough length per animal to ensure access to feed.
(18)	The holding pens are overcrowded, thereby not providing enough space for every animal to lie down or stand up without hindrance.
(19)	The holding area or unloading area is not properly weather-proofed, resulting in animals being exposed to adverse weather conditions.
(20)	Failure to provide sufficient bedding or any bedding in the holding area.
(21)	Deliberately throwing, dropping or knocking over a container with live animals.
(22)	Animals of different species, sexes, ages, or origins are housed together (despite clearly being incompatible) which may lead to fights, injuries, and unnecessary stress.
(23)	Tying an animal’s legs together or to a post in violation of animal welfare standards.
(24)	Lactating dairy cattle are not milked at least every twelve hours.
(25)	Dragging animals that are too weak or injured to walk on their own using painful tools such as a winch or other driving aids.
(26)	Sick or injured animals (which are obviously in severe pain, have large or deep wounds, are bleeding severely, or show a severely disturbed general condition) are housed with healthy animals in holding pens instead of being prioritised for immediate slaughter or euthanasia.
(27)	Inadequate immobilisation of an animal prior to stunning (e.g., the animal can turn around in the stun box/no head restraint when immobilising solipeds or cattle).
(28)	Prohibited methods of immobilizing animals, such as using a bolt shot to the neck.
(29)	Using a stunning device with visible defects (such as corroded electrodes, a bent bolt, or worn buffer rubbers) to stun an animal.
(30)	The absence of proper spare equipment or replacement parts for worn components of bolt gun equipment (e.g., buffer rubbers, recuperating spring) during stunning.
(31)	The use of outdated or old stunning equipment that does not meet current animal welfare standards.
(32)	The water bath stunning equipment is inadequate.
(33)	The attachment of the stunning device is incorrect (e.g., the bolt firing device is not positioned on the head correctly, such as not being vertical or secure, causing the bolt to be fired incorrectly or not making contact).
(34)	Failure to perform an assessment of the effectiveness of stunning.
(35)	An animal is slaughtered without prior stunning, without a respective official exemption permit (e.g., for the purpose of Halal/Kosher slaughter)
(36)	An animal that has been ineffectively stunned (e.g., one that shows signs of consciousness such as spontaneous blinking, directed eye movements or reactions to touch) is not re-stunned before exsanguination.
(37)	The time limit allowed between stunning and bleeding is exceeded without a certificate of exemption (in accordance with Section 13(2) of the German Ordinance on the Protection of Animals during Slaughter or Killing).
(38)	Further preparation or scalding of slaughtered animals (such as removing the head, eyes, or ears) is performed while the animal is still showing signs of consciousness.
(39)	Handling and caring for animals prior to stunning is performed by an unqualified person without the required certificate of competence.
(40)	Stunning, killing, and related tasks are carried out by an individual without the required certification of competence.

**Table 2 animals-13-02916-t002:** Number of different abattoirs in which the OVs who participated in the online survey worked in the past three years (N = 108 online survey participants per category).

Number of Different Abattoirs	% of N = 108
1–3	61.8
4–9	33.0
10 or more	5.2

**Table 3 animals-13-02916-t003:** Percentage of OVs participating in the online survey who monitored at least one abattoir in the respective categories of slaughtered livestock units (LU) per week (N = 108 online survey participants per category).

Number of LU Slaughtered per Week	% of N
<20	68.5
20 to 100	40.1
>100	59.8

**Table 4 animals-13-02916-t004:** Percentage of OVs participating in the online survey who reportedly had experience in overseeing the slaughter of the different species listed below (N = 108 online survey participants per category).

Species	% of N
Pigs	87.0
Cattle	84.3
Sheep	57.4
Goats	42.6
Poultry	26.9
Horses	13.9
Other animals	10.1

**Table 5 animals-13-02916-t005:** Percentage of OVs participating in the online survey who reportedly had experience in overseeing different stunning methods (N = 108 online survey participants per category).

Stunning Method	% of N
Captive bolt stunning	90.5
Electrical stunning	83.8
Carbon dioxide stunning	20.0
Electrical water-bath stunning for poultry	16.2

## Data Availability

Data sharing not applicable.

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
