# Peer review of "Measures and Penalties for Animal Welfare Violations at German Abattoirs: A Compilation of Current Recommendations and Practices"

_animals, 2023, doi:10.3390/ani13182916_

Round 1

Reviewer 1 Report

Dear authors, what a refreshing piece of science you delivered! This paper was a joy to read. Innovative in its approach, thoroughly researched, well-written, and most of all, one that could have a concrete impact on the welfare of those animals that are faced with their final hours of life. Thank you for looking into such important matters. I do hope you will take this strand of research further. The research community can and should have a more prominent role in advising the competent authorities as to how to better enforce existing animal welfare laws. Official inspectors are, indeed, often left alone facing threats or, quite simply, a lack of feedback on their work. All this needs to get out there, and thanks to you, it will. Chapeau. With my warmest regards, your reviewer.

I have a couple of comments that you may wish to take into account. 

Introduction

I think that the introduction would benefit from a slight reorganisation of the logical arguments. I would say, first the existing rules and regulations (EU and German) that apply to abattoir operations; second, the fact that literature and undercover investigations show that there are sometimes severe violations; third, the fact that enforcement appears to be patchy and that, even when it happens, the sanctions are not dissuasive as there is a certain degree of tolerance towards mistreatment of farmed animals.

L 58-61 This is an abrupt transition. Please clarify the relevance of media attention on undercover investigations. Perhaps a transition could be that what is reported in the literature is confirmed by the undercover investigations, which in turn get a lot of media attention.

L 272 For added clarity, I would include in this section a table with the 40 violations identified and discussed by the OVs.  

L 114-117 Indeed a very important aspect!

Supplementary materials: 

Please change seite (-> page) everywhere in the document

Could you kindly define daily rates? They are present in the last column (on the right, Judicial Decisions/notes)

Author Response

Dear Sir or Madam,

We would like to express our gratitude for your thoughtful review and valuable feedback on the article entitled 'Measures and Penalties for Animal Welfare Violations at German Abattoirs: A Compilation of Current Recommendations and Practices.'

Your insightful comments have been instrumental in identifying areas for improvement, and we sincerely appreciate the time and effort you dedicated to reviewing our work. Your positive assessment of the overall quality of our article is encouraging, and we are committed to making the necessary modifications to enhance its clarity and impact.

We incorporated all the suggested changes. Our responses to your comments are as follows:

Point 1: “I think that the introduction would benefit from a slight reorganisation of the logical arguments. I would say, first the existing rules and regulations (EU and German) that apply to abattoir operations; second, the fact that literature and undercover investigations show that there are sometimes severe violations; third, the fact that enforcement appears to be patchy and that, even when it happens, the sanctions are not dissuasive as there is a certain degree of tolerance towards mistreatment of farmed animals.”

Response 1: Thank you for this instructive suggestion, we changed the text accordingly.

Point 2: L 58-61 This is an abrupt transition. Please clarify the relevance of media attention on undercover investigations. Perhaps a transition could be that what is reported in the literature is confirmed by the undercover investigations, which in turn get a lot of media attention.

Response 2: We modified the text accordingly: “The findings reported in literature are corroborated by undercover investigations by animal welfare organisations, which receive a lot of media attention [15]. Articles describing (and sometimes showing) footage of violations of Regulation (EC) No 1099/2009 frequently make headlines in European media (e.g., beating animals or slaughtering ineffectively stunned animals) [16-20]. Additionally, publications describe that there is a deficit in the enforcement of animal welfare laws and regulations at German abattoirs [21].”

Point 3: L 272 For added clarity, I would include in this section a table with the 40 violations identified and discussed by the OVs. 

Response 3: This suggestion was incorporated as Table 1.

Point 4: L 114-117 Indeed a very important aspect!

Response 4: Thank you, we appreciate your enthusiasm for our project.

Supplementary material 1:

Point 5: Please change seite (-> page) everywhere in the document

Response 5: Thank you for pointing this out, we changed it accordingly.

Point 6: Could you kindly define daily rates? They are present in the last column (on the right, Judicial Decisions/notes)

Response 6: A definition was included on page 6 of supplementary material 1 as follows: “Section 40 of the German Criminal Code describes how fines are imposed: A fine is imposed in daily rates. The minimum fine is five and, unless otherwise provided by law, the maximum is 360 full daily rates. The court determines the amount of the daily rate having regard to the offender’s personal and financial circumstances. In doing so, it typically bases its assessment on the average net income which the offender earns or could earn in one day. A daily rate is set at no less than 1 euro and no more than 30,000 euros. The offender’s income and assets and other relevant assessment factors may be estimated when setting the amount of the daily rate. The number and amount of the daily rates are indicated in the decision.”

In the file attached, you can find the updated manuscript.

Best wishes,

Stephanie Schneidewind, Prof. Diana Meemken, Dr. Susann Langforth

Reviewer 2 Report

The authors performed an interesting and novel study.

I would suggest to publish the survey and try to do some extra statistical analysis a part from the descriptive statistics.

Try to represent better the results and interpreter them, without repeating them, using a bit more literature in the discussion

For the survey it would be important to understand the target population, any OVs or OVs who conducted at least one inspection at a slaughterhouses. Can you add a power analysis to understand whether 66 completed survey is significant based on the target population?

306-309, do not repeat your aim in the results

316-326: Can those results be put in a table?

Avoid to put sub title in the discussion

Author Response

Dear Sir or Madam,

We would like to thank you for your thoughtful review and critical feedback on the article entitled 'Measures and Penalties for Animal Welfare Violations at German Abattoirs: A Compilation of Current Recommendations and Practices.'

Your feedback has played an important role in pinpointing areas for improvement, and we value the time you invested in evaluating our efforts. We are committed to making the necessary modifications to improve this article based on your suggestions.

Our responses to your comments are as follows:

Point 1: I would suggest to publish the survey and try to do some extra statistical analysis a part from the descriptive statistics.

Response 1: We will publish the survey in English and German language as supplementary materials 2 and 3 (supplementary materials 2 = our survey in English; supplementary materials 3 = our survey in German). As for further statistical analysis, we considered the following ideas and will describe the respective challenges. We considered investigating the potential correlation between the severity of an animal welfare violation and the measures recommended in the online survey. However, implementing a test would necessitate classifying the violations into discrete categories such as "mild," "moderate," and "severe," or at least "mild" and "severe." Regrettably, such categorization is presently unfeasible if we intend to do so objectively. We conducted some research and found no categorization of different violations against German laws. We considered categorizing our violations according to how other countries categorize these violations (i.e. Canada: https://laws-lois.justice.gc.ca/eng/regulations/SOR-2000-187/page-2.html#docCont ), however, not all of the violations we included in our project are addressed. This is underscored by the results of the online survey. Our online survey participants, all of whom were actively involved monitoring compliance to animal welfare laws in abattoirs, displayed significant variability in their suggestions for appropriate measures concerning specific animal welfare violations. This divergence in responses reflects the diverse perspectives they hold regarding the severity of individual cases. Furthermore, it must be said that the recommendations listed in supplementary materials 1 are not just based on the OVs who participated in this survey. The OVs who participated in the virtual colloquium also significantly contributed to the compiled recommendations. However, we intentionally did not ask the participants in the virtual colloquium for information on the abattoirs they monitored to protect their anonymity. Thus, doing statistical analysis only based on the recommendations of the participants would only be representative for a fraction of the persons who contributed to our list of recommendations.

We also explored the possibility of conducting a test to investigate any potential correlation between the size of abattoirs or the species they were most experienced with monitoring, and the severity of the measures recommended. However, upon closer examination of the data, we observed that the majority of participants had monitored a diverse range of abattoirs in terms of size and species slaughtered over the past three years. Also, this model would require us to categorize the possible measures, which would also have to be based on our subjective interpretation of the laws (we conducted some research and found no categorizing of different measures applicable in the abattoir).

If you have a different perspective on this matter, we would greatly appreciate it if you could kindly suggest a specific statistical model that you believe might be appropriate.

Point 2: Try to represent better the results and interpreter them, without repeating them, using a bit more literature in the discussion

Response 2: The discussion section was restructured and now includes the following passage concerning the interpretation of the results: “As for interpreting the results of this study, lawyers and OV with experience in law enforcement in abattoirs should carefully review the list and identify whether the measures and fines suggested are adequate. However, the results of the online-survey on their own suggest that animal protection laws at the abattoir are under-enforced by some OV. Most of the violations were at least regulatory offences, but the average amount of par-ticipants suggesting an initiation of a regulatory offence proceeding after the first violation was only 18.9%. For the repeated violation, this percentage increased to 45.5%. We would have expected the percentage of participants recommending regulatory fines in the viola-tions depicted in Figures 1 and 2 to be closer to 100.0%, seeing as Section 16 lists this vio-lation as a regulatory offence which should be fined. Additionally, the results of the online survey showed that many OV would not file a criminal complaint in cases of significant pain or prolonged suffering. For example, many OV suggested that dragging a downer cattle off of the transportation vehicle instead of humanely killing it on the spot is a regu-latory offence, even though this should be prosecuted as a crime due to the significant and prolonged pain it inflicts on the animal. Perhaps this is due to the lack of support many OV reportedly receive from the department head of their office, or these results suggest that OV may need further training in identifying animal welfare violations and implementing the measures and fines the law requires them to. However, it is also important to note that the results of the survey contrasted the results of the virtual colloquium. During the virtual colloquium, participants suggested initiating measures aimed at addressing the specific underlying issues causing an animal welfare violation. This often involved an animal welfare order based on Section 16a of the German Animal Welfare Act. This creates the impression that the animal welfare order based on Section 16a is a more promising solu-tion than punishment for less severe cases. This approach focuses on solving problems rather than punishing abattoir employees and/or operators. As for the fines listed in judi-cial decisions, it is known that most public prosecutors have little experience with animal welfare violations of farm animals and thus may suggest rather low fines [25]. However, on the other hand, some lawyers would suggest very high sanctions, as demonstrated by Jens Bülte’s suggestion to increase the prison sentence for crimes committed against ani-mals from three to five years [39]. When penalising a violation, the individual level of pain, suffering and/or harm inflicted on the animal(s) must be considered. Moreover, the income and personal circumstances, in addition to whether it was their first or repeated violation, must be taken into account. Also, a decision must be made as to whether the general behaviour of the persons accused indicates empathy for the animals affected, or whether animals merely have the status of goods. This may, to some extent, explain why the participants’ responses regarding appropriate measures and penalties for specific animal welfare cases varied so much in the survey (as exemplified by Figures 1 and 2). Further reasons, such as the lack of guidelines, are likely to contribute to the heterogeneity of participants’ assessments. Whether the regulatory fines and/or periodic penalty pay-ments proposed by participants are too high, too low or just right, is rather subjective and might depend on the personal experiences gained in the context of a similar past case which occurred in an OV working environment. Therefore, these measures and penalties cannot be used as guidelines for law enforcement at abattoirs, but rather provide insights in the status quo and provides a basis for developing clear guidelines.”

Point 3: For the survey it would be important to understand the target population, any OVs or OVs who conducted at least one inspection at an abattoir. Can you add a power analysis to understand whether 66 completed survey is significant based on the target population?

Response 3: We considered the idea of adding a power analysis to understand whether the 66 persons who completed the entire survey are representative. However, we do not have any knowledge about the number of official veterinarians monitoring animal welfare in abattoirs. We reached out to the German Federal Association of Civil Servant Veterinarians (Bundesverband der beamteten Tierärzte e. V.), inquiring whether they could provide us this information. Unfortunately, their response was negative. Published statistics include the number of official veterinarians in Germany, but many of these official veterinarians are currently not working in abattoirs, seeing as many larger cities do not have an abattoir and OV do not monitor outside of the district/town they are employed in. Furthermore, as described in Response 1, it is important to note that the 66 persons who completed the entire survey are not representative for the persons who contributed to the completion of this project. The OV who participated in the online virtual colloquium are not included, but they significantly contributed to the list of measures and penalties provided in supplementary materials 1. It is unclear whether some of the survey-participants also participated in the virtual colloquium, so determining the number of persons who contributed to this compilation is not possible.

Point 4: 306-309, do not repeat your aim in the results

Response 4: Thank you for pointing this out. The following text was deleted from the article: “The aim of the survey was to gather information on measures and penalties taken in past animal welfare cases as well as to measures and penalties participants considered appropriate (hypothetically).”

Point 5: 316-326: Can those results be put in a table?

Response 5: Thank you for this suggestion. The results were put in Tables 2 - 5.

Point 6: Avoid to put sub title in the discussion

Response 6: The sub titles were removed from the discussion section.

In the file attached, you can find the updated manuscript.

Thank you again for your input.

Best wishes,

Stephanie Schneidewind, Prof. Diana Meemken, Dr. Susann Langforth
